# Accelerating Stochastic Composition Optimization

**Mengdi Wang**∗, **Ji Liu**∗, **and Ethan X. Fang**
Princeton University, University of Rochester, Pennsylvania State University
mengdiw@princeton.edu, ji.liu.uwisc@gmail.com, xxf13@psu.edu

## Abstract

Consider the stochastic composition optimization problem where the objective is a composition of two expected-value functions. We propose a new stochastic first-order method, namely the accelerated stochastic compositional proximal gradient (ASC-PG) method, which updates based on queries to the sampling oracle using two different timescales. The ASC-PG is the first proximal gradient method for the stochastic composition problem that can deal with nonsmooth regularization penalty. We show that the ASC-PG exhibits faster convergence than the best known algorithms, and that it achieves the optimal sample-error complexity in several important special cases. We further demonstrate the application of ASC-PG to reinforcement learning and conduct numerical experiments.

## 1 Introduction

The popular stochastic gradient methods are well suited for minimizing expected-value objective functions or the sum of a large number of loss functions. Stochastic gradient methods find wide applications in estimation, online learning, and training of deep neural networks. Despite their popularity, they do not apply to the minimization of a nonlinear function involving expected values or a composition between two expected-value functions.

In this paper, we consider the *stochastic composition problem*, given by

$$\min_{\mathbf{x} \in \Re^n} \quad H(\mathbf{x}) := \underbrace{\mathbb{E}_v \big( f_v \big( \mathbb{E}_w (g_w(\mathbf{x})) \big) \big)}_{=:F(\mathbf{x})} + R(\mathbf{x}) \tag{1}$$

where $(f \circ g)(\mathbf{x}) = f(g(\mathbf{x}))$ denotes the function composition, $g_w(\cdot) : \Re^n \mapsto \Re^m$ and $f_v(\cdot) : \Re^m \mapsto \Re$ are continuously differentiable functions, $v, w$ are random variables, and $R(\mathbf{x}) : \Re^n \mapsto \Re \cup \{+\infty\}$ is an extended real-valued closed convex function. We assume throughout that there exists at least one optimal solution $\mathbf{x}^*$ to problem (1). We focus on the case where $f_v$ and $g_w$ are smooth, but we allow $R$ to be a nonsmooth penalty such as the $\ell_1$-norm. We do no require either the outer function $f_v$ or the inner function $g_w$ to be convex or monotone. As a result, the composition problem cannot be reformulated into a saddle point problem in general.

Our algorithmic objective is to develop efficient algorithms for solving problem (1) based on random evaluations of $f_v$, $g_w$ and their gradients. Our theoretical objective is to analyze the rate of convergence for the stochastic algorithm and to improve it when possible. In the online setting, the iteration complexity of our stochastic methods can be interpreted as a sample-error complexity upper bound for estimating the optimal solution of problem (1).

### 1.1 Motivating Examples

One motivating example is *reinforcement learning* [Sutton and Barto, 1998]. Consider a controllable Markov chain with states $1, \ldots, S$. Estimating the value-per-state of a fixed control policy $\pi$ is known

---

∗Equal contribution.

as on-policy learning. It can be casted into an $S \times S$ system of Bellman equations:

$$\gamma P^{\pi} V^{\pi} + r^{\pi} = V^{\pi},$$

where $\gamma \in (0, 1)$ is a discount factor, $P_{s\tilde{s}}^{\pi}$ is the transition probability from state $s$ to state $\tilde{s}$, and $r_s^{\pi}$ is the expected state transition reward at state $s$. The solution $V^{\pi}$ to the Bellman equation is the value vector, with $V^{\pi}(s)$ being the total expected reward starting at state $s$. In the blackbox simulation environment, $P^{\pi}, r^{\pi}$ are unknown but can be sampled from a simulator. As a result, solving the Bellman equation becomes a special case of the stochastic composition optimization problem:

$$\min_{\mathbf{x} \in \Re^S} \quad \|\mathbb{E}[A]\mathbf{x} - \mathbb{E}[\mathbf{b}]\|^2, \tag{2}$$

where $A, \mathbf{b}$ are random matrices and random vectors such that $\mathbb{E}[A] = I - \gamma P^{\pi}$ and $\mathbb{E}[\mathbf{b}] = r^{\pi}$. It can be viewed as the composition of the square norm function and the expected linear function. We will give more details on the reinforcement learning application in Section 4.

Another motivating example is *risk-averse learning*. For example, consider the mean-variance minimization problem

$$\min_{\mathbf{x}} \quad \mathbb{E}_{a,b}[h(\mathbf{x}; a, b)] + \lambda \mathrm{Var}_{a,b}[h(\mathbf{x}; a, b)],$$

where $h(x; a, b)$ is some loss function parameterized by random variables $a$ and $b$, and $\lambda > 0$ is a regularization parameter. Its batch version takes the form

$$\min_{\mathbf{x}} \quad \frac{1}{N} \sum_{i=1}^{N} h(\mathbf{x}; a_i, b_i) + \frac{\lambda}{N} \sum_{i=1}^{N} \left( h(\mathbf{x}; a_i, b_i) - \frac{1}{N} \sum_{i=1}^{N} h(\mathbf{x}; a_i, b_i) \right)^2.$$

Here the variance term is the composition of the mean square function and an expected loss function. Although the stochastic composition problem (1) was barely studied, it actually finds a broad spectrum of emerging applications in estimation and machine learning (see Wang et al. [2016] for a list of applications). Fast optimization algorithms with theoretical guarantees will lead to new computation tools and online learning methods for a broader problem class, no longer limited to the expectation minimization problem.

## 1.2 Related Works and Contributions

Contrary to the expectation minimization problem, "unbiased" gradient samples are no longer available for the stochastic composition problem (1). The objective is *nonlinear* in the joint probability distribution of $(w, v)$, which substantially complicates the problem. In a recent work by Dentcheva et al. [2015], a special case of the stochastic composition problem, i.e., risk-averse optimization, has been studied. A central limit theorem has been established, showing that the $K$-sample batch problem converges to the true problem at the rate of $O(1/\sqrt{K})$ in a proper sense. For the case where $R(x) = 0$, Wang et al. [2016] has proposed and analyzed a class of *stochastic compositional gradient/subgradient methods* (SCGD). The SCGD involves two iterations of different time scales, one for estimating $x^*$ by a stochastic quasi-gradient iteration, the other for maintaining a running estimate of $g(x^*)$. Wang and Liu [2016] studies the SCGD in the setting where samples are corrupted with Markov noises (instead of i.i.d. zero-mean noises). Both works establish almost sure convergence of the algorithm and several convergence rate results, which are the best-known convergence rate prior to the current paper.

The idea of using two-timescale quasi-gradient traced back to the earlier work Ermoliev [1976]. The incremental treatment of proximal gradient iteration has been studied extensively for the expectation minimization problem, see for examples Beck and Teboulle [2009], Bertsekas [2011], Ghadimi and Lan [2015], Gurbuzbalaban et al. [2015], Nedić [2011], Nedić and Bertsekas [2001], Nemirovski et al. [2009], Rakhlin et al. [2012], Shamir and Zhang [2013], Wang and Bertsekas [2016], Wang et al. [2015]. However, except for Wang et al. [2016] and Wang and Liu [2016], all of these works focus on variants of the expectation minimization problem and do not apply to the stochastic composition problem (1). Another work partially related to this paper is by Dai et al. [2016]. They consider a special case of problem (1) arising in kernel estimation, where they assume that all functions $f_v$'s are convex and their conjugate functions $f_v^{\star}$'s can be easily obtained/sampled. Under these additional assumptions, they essentially rewrite the problem into a saddle point optimization involving functional variables.

In this paper, we propose a new accelerated stochastic compositional proximal gradient (ASC-PG) method that applies to the penalized problem (1), which is a more general problem than the one considered in Wang et al. [2016]. We use a *coupled martingale stochastic analysis* to show that ASC-PG achieves significantly better sample-error complexity in various cases. We also show that ASC-PG exhibits optimal sample-error complexity in two important special cases: the case where the outer function is linear and the case where the inner function is linear.

Our contributions are summarized as follows:

1. We propose the first stochastic *proximal-gradient* method for the stochastic composition problem. This is the first algorithm that is able to address the nonsmooth regularization penalty $R(\cdot)$ without deteriorating the convergence rate.
2. We obtain a convergence rate $O(K^{-4/9})$ for smooth optimization problems that are not necessarily convex, where $K$ is the number of queries to the stochastic first-order oracle. This improves the best known convergence rate and provides a new benchmark for the stochastic composition problem.
3. We provide a comprehensive analysis and results that apply to various special cases. In particular, our results contain as special cases the known optimal rate results for the expectation minimization problem, i.e., $O(1/\sqrt{K})$ for general objectives and $O(1/K)$ for strongly convex objectives.
4. In the special case where the inner function $g(\cdot)$ is a linear mapping, we show that it is sufficient to use one timescale to guarantee convergence. Our result achieves the non-improvable rate of convergence $O(1/K)$ for optimal strongly convex optimization and $O(1/\sqrt{K})$ for nonconvex smooth optimization. It implies that the inner linearity does not bring fundamental difficulty to the stochastic composition problem.
5. We show that the proposed method leads to a new on-policy reinforcement learning algorithm. The new learning algorithm achieves the optimal convergence rate $O(1/\sqrt{K})$ for solving Bellman equations (or $O(1/K)$ for solving the least square problem) based on $K$ observations of state-to-state transitions.

In comparison with Wang et al. [2016], our analysis is more succinct and leads to stronger results. To the best of our knowledge, Theorems 1 and 2 in this paper provide the best-known rates for the stochastic composition problem.

**Paper Organization.**   Section 2 states the sampling oracle and the accelerated stochastic compositional proximal gradient algorithm (ASC-PG). Section 3 states the convergence rate results in the case of general nonconvex objective and in the case of strongly convex objective, respectively. Section 4 describes an application of ASC-PG to reinforcement learning and gives numerical experiments.

**Notations and Definitions.**   For $\mathbf{x} \in \Re^n$, we denote by $\mathbf{x}'$ its transpose, and by $\|\mathbf{x}\|$ its Euclidean norm (i.e., $\|\mathbf{x}\| = \sqrt{\mathbf{x}'\mathbf{x}}$). For two sequences $\{\mathbf{y}_k\}$ and $\{\mathbf{z}_k\}$, we write $\mathbf{y}_k = O(\mathbf{z}_k)$ if there exists a constant $c > 0$ such that $\|\mathbf{y}_k\| \le c\|\mathbf{z}_k\|$ for each $k$. We denote by $\mathbf{I}^{\mathrm{value}}_{\mathrm{condition}}$ the indicator function, which returns "value" if the "condition" is satisfied; otherwise 0. We denote by $H^*$ the optimal objective function value of problem (1), denote by $X^*$ the set of optimal solutions, and denote by $\mathcal{P}_S(\mathbf{x})$ the Euclidean projection of $\mathbf{x}$ onto $S$ for any convex set $S$. We also denote by short that $f(\mathbf{y}) = \mathbb{E}_v[f_v(\mathbf{y})]$ and $g(\mathbf{x}) = \mathbb{E}_w[g_w(\mathbf{x})]$.

## 2   Algorithm

We focus on the black-box sampling environment. Suppose that we have access to a stochastic first-order oracle, which returns random realizations of first-order information upon queries. This is a typical simulation oracle that is available in both online and batch learning. More specifically, assume that we are given a **Sampling Oracle (SO)** such that

- Given some $\mathbf{x} \in \Re^n$, the **SO** returns a random vector $g_w(x)$ and a noisy subgradient $\nabla g_w(\mathbf{x})$.
- Given some $\mathbf{y} \in \Re^m$, the **SO** returns a noisy gradient $\nabla f_v(\mathbf{y})$.

Now we propose the Accelerated Stochastic Compositional Proximal Gradient (ASC-PG) algorithm, see Algorithm 1. ASC-PG is a generalization of the SCGD proposed by Wang et al. [2016], in which a proximal step is used to replace the projection step.

In Algorithm 1, the *extrapolation-smoothing scheme* (i.e., the $(\mathbf{y}, \mathbf{z})$-step) is critical to the acceleration of convergence. The acceleration is due to the fast running estimation of the unknown quantity

---

**Algorithm 1** Accelerated Stochastic Compositional Proximal Gradient (ASC-PG)

---

**Require:** $\mathbf{x}_1 \in \Re^n$, $\mathbf{y}_0 \in \Re^m$, **SO**, $K$, stepsize sequences $\{\alpha_k\}_{k=1}^K$, and $\{\beta_k\}_{k=1}^K$.
**Ensure:** $\{\mathbf{x}_k\}_{k=1}^K$
1: Initialize $\mathbf{z}_1 = \mathbf{x}_1$.
2: **for** $k = 1, \cdots, K$ **do**
3:    Query the **SO** and obtain gradient samples $\nabla f_{v_k}(\mathbf{y}_k)$, $\nabla g_{w_k}(\mathbf{z}_k)$.
4:    Update the main iterate by

$$\mathbf{x}_{k+1} \quad = \quad \mathrm{prox}_{\alpha_k R(\cdot)} \left( \mathbf{x}_k - \alpha_k \nabla g_{w_k}^\top(\mathbf{x}_k) \nabla f_{v_k}(\mathbf{y}_k) \right).$$

5:    Update auxillary iterates by an *extrapolation-smoothing* scheme:

$$\mathbf{z}_{k+1} \quad = \quad \left( 1 - \frac{1}{\beta_k} \right) \mathbf{x}_k + \frac{1}{\beta_k} \mathbf{x}_{k+1},$$

$$\mathbf{y}_{k+1} \quad = \quad (1 - \beta_k)\mathbf{y}_k + \beta_k g_{w_{k+1}}(\mathbf{z}_{k+1}),$$

   where the sample $g_{w_{k+1}}(\mathbf{z}_{k+1})$ is obtained via querying the **SO**.
6: **end for**

---

$g(\mathbf{x}_k) := \mathbb{E}_w[g_w(\mathbf{x}_k)]$. At iteration $k$, the running estimate $\mathbf{y}_k$ of $g(\mathbf{x}_k)$ is obtained using a weighted smoothing scheme, corresponding to the $\mathbf{y}$-step; while the new query point $\mathbf{z}_{k+1}$ is obtained through extrapolation, corresponding to the $\mathbf{z}$-step. The updates are constructed in a way such that $\mathbf{y}_k$ is a nearly unbiased estimate of $g(\mathbf{x}_k)$. To see how the extrapolation-smoothing scheme works, we let the weights be

$$\xi_t^{(k)} = \begin{cases} \beta_t \prod_{i=t+1}^k (1 - \beta_i), & \text{if } k > t \geq 0 \\ \beta_k, & \text{if } k = t \geq 0. \end{cases} \tag{3}$$

Then we can verify the following important relations:

$$\mathbf{x}_{k+1} = \sum_{t=0}^k \xi_t^{(k)} \mathbf{z}_{t+1}, \qquad \mathbf{y}_{k+1} = \sum_{t=0}^k \xi_t^{(k)} g_{w_{t+1}}(\mathbf{z}_{t+1}),$$

which essentially say that $\mathbf{x}_{k+1}$ is a damped weighted average of $\{z_{t+1}\}_0^{k+1}$ and $\mathbf{y}_{k+1}$ is a damped weighted average of $\{g_{w_{t+1}}(z_{t+1})\}_0^{k+1}$.

**An Analytical Example of the Extrapolation-Smooth Scheme**   Now consider the special case where $g_w(\cdot)$ is always a linear mapping $g_w(z) = A_w z + b_z$ and $\beta_k = 1/(k+1)$. We can verify that $\xi_t^{(k)} = 1/(k+1)$ for all $t$. Then we have

$$g(\mathbf{x}_{k+1}) = \frac{1}{k+1} \sum_{t=0}^k \mathbb{E}[A_w]\mathbf{z}_{t+1} + \mathbb{E}[\mathbf{b}_w], \qquad \mathbf{y}_{k+1} = \frac{1}{k+1} \sum_{t=0}^k A_{w_{t+1}} \mathbf{z}_{t+1} + \frac{1}{k+1} \sum_{t=0}^k \mathbf{b}_{w_{t+1}}.$$

In this way, we can see that the scaled error

$$k(\mathbf{y}_{k+1} - g(\mathbf{x}_{k+1})) = \sum_{t=0}^k (A_{w_{t+1}} - \mathbb{E}[A_w])\mathbf{z}_{t+1} + \sum_{t=0}^k (\mathbf{b}_{w_{t+1}} - \mathbb{E}[\mathbf{b}_w])$$

is a *zero-mean* and *zero-drift* martingale. Under additional technical assumptions, we have

$$\mathbb{E}[\|\mathbf{y}_{k+1} - g(\mathbf{x}_{k+1})\|^2] \leq O(1/k).$$

Note that the zero-drift property of the error martingale is the key to the fast convergence rate. The zero-drift property comes from the near-unbiasedness of $\mathbf{y}_k$, which is due to the special construction of the extrapolation-smoothing scheme. In the more general case where $g_w$ is not necessarily linear, we can use a similar argument to show that $\mathbf{y}_k$ is a nearly unbiased estimate of $g(\mathbf{x}_k)$. As a result, the extrapolation-smoothing $(\mathbf{y}, \mathbf{z})$-step ensures that $\mathbf{y}_k$ tracks the unknown quantity $g(\mathbf{x}_k)$ efficiently.

# 3 Main Results

We present our main theoretical results in this section. Let us begin by stating our assumptions. Note that all assumptions involving random realizations of $v, w$ hold with probability 1.

**Assumption 1.** *The samples generated by the* **SO** *are unbiased in the following sense:*

1. $\mathbb{E}_{\{w_k, v_k\}}(\nabla g_{w_k}^\top(\mathbf{x}) \nabla f_{v_k}(\mathbf{y})) = \nabla g^\top(\mathbf{x}) \nabla f(\mathbf{y}) \quad \forall k = 1, 2, \cdots, K, \quad \forall \mathbf{x}, \forall \mathbf{y}.$

2. $\mathbb{E}_{w_k}(g_{w_k}(\mathbf{x})) = g(\mathbf{x}) \quad \forall \mathbf{x}.$

*Note that $w_k$ and $v_k$ are not necessarily independent.*

**Assumption 2.** *The sample gradients and values generated by the* **SO** *satisfy*

$$\mathbb{E}_w(\|g_w(\mathbf{x}) - g(\mathbf{x})\|^2) \le \sigma^2 \quad \forall \mathbf{x}.$$

**Assumption 3.** *The sample gradients generated by the* **SO** *are uniformly bounded, and the penalty function $R$ has bounded gradients.*

$$\|\nabla f_v(\mathbf{x})\| \le \Theta(1), \quad \|\nabla g_w(\mathbf{x})\| \le \Theta(1), \quad \|\partial R(\mathbf{x})\| \le \Theta(1) \quad \forall \mathbf{x}, \forall w, \forall v$$

**Assumption 4.** *There exist $L_F, L_f, L_g > 0$ such that*

1. $F(\mathbf{z}) - F(\mathbf{x}) \le \langle \nabla F(\mathbf{x}), \mathbf{z} - \mathbf{x} \rangle + \frac{L_F}{2} \|\mathbf{z} - \mathbf{x}\|^2 \quad \forall \mathbf{x} \, \forall \mathbf{z}.$

2. $\|\nabla f_v(\mathbf{y}) - \nabla f_v(\mathbf{w})\| \le L_f \|\mathbf{y} - \mathbf{w}\| \quad \forall \mathbf{y} \, \forall \mathbf{w} \, \forall v.$

3. $\|g(\mathbf{x}) - g(\mathbf{z}) - \nabla g(\mathbf{z})^\top(\mathbf{x} - \mathbf{z})\| \le \frac{L_g}{2} \|\mathbf{x} - \mathbf{z}\|^2 \quad \forall \mathbf{x} \, \forall \mathbf{z}.$

Our first main result concerns with general optimization problems which are not necessarily convex.

**Theorem 1** (Smooth (Nonconvex) Optimization)**.** *Let Assumptions 1, 2, 3, and 4 hold. Denote by $F(\mathbf{x}) := (\mathbb{E}_v(f_v) \circ \mathbb{E}_w(g_w))(\mathbf{x})$ for short and suppose that $R(\mathbf{x}) = 0$ in (1) and $\mathbb{E}(F(\mathbf{x}_k))$ is bounded from above. Choose $\alpha_k = k^{-a}$ and $\beta_k = 2k^{-b}$ where $a \in (0,1)$ and $b \in (0,1)$ in Algorithm 1. Then we have*

$$\frac{\sum_{k=1}^K \mathbb{E}(\|\nabla F(\mathbf{x}_k)\|^2)}{K} \le O(K^{a-1} + L_f^2 L_g K^{4b-4a} \mathbf{I}_{4a-4b=1}^{\log K} + L_f^2 K^{-b} + K^{-a}). \tag{4}$$

*If $L_g \ne 0$ and $L_f \ne 0$, choose $a = 5/9$ and $b = 4/9$, yielding*

$$\frac{1}{K} \sum_{k=1}^K \mathbb{E}(\|\nabla F(\mathbf{x}_k)\|^2) \le O(K^{-4/9}). \tag{5}$$

*If $L_g = 0$ or $L_f = 0$, choose $a = b = 1/2$, yielding*

$$\frac{1}{K} \sum_{k=1}^K \mathbb{E}(\|\nabla F(\mathbf{x}_k)\|^2) \le O(K^{-1/2}). \tag{6}$$

The result of Theorem 1 strictly improves the best-known results given by Wang et al. [2016]. First the result of (5) improves the finite-sample error bound from $O(k^{-2/7})$ to $O(k^{-4/9})$ for general convex and nonconvex optimization. This improves the best known convergence rate and provides a new benchmark for the stochastic composition problem. Note that it is possible to relax the condition "$\mathbb{E}(F(\mathbf{x}_k))$ is bounded from above" in Theorem 1. However, it would make the analysis more cumbersome and yield an additional term $\log K$ in the error bound.

Our second main result concerns strongly convex objective functions. We say that the objective function $H$ is *optimally strongly convex* with parameter $\lambda > 0$ if

$$H(\mathbf{x}) - H(\mathcal{P}_{X^*}(\mathbf{x})) \ge \lambda \|\mathbf{x} - \mathcal{P}_{X^*}(\mathbf{x})\|^2 \quad \forall \mathbf{x}. \tag{7}$$

(see Liu and Wright [2015]). Note that any strongly convex function is optimally strongly convex, but the reverse does not hold. For example, the objective function (2) in on-policy reinforcement learning is always optimally strongly convex (even if $\mathbb{E}(A)$ is a rank deficient matrix), but not necessarily strongly convex.

**Theorem 2.** *(Strongly Convex Optimization) Suppose that the objective function $H(\mathbf{x})$ in (1) is optimally strongly convex with parameter $\lambda > 0$ defined in (7). Set $\alpha_k = C_a k^{-a}$ and $\beta_k = C_b k^{-b}$ where $C_a > 4\lambda$, $C_b > 2$, $a \in (0, 1]$, and $b \in (0, 1]$ in Algorithm 1. Under Assumptions 1, 2, 3, and 4, we have*

$$\mathbb{E}(\|\mathbf{x}_K - \mathcal{P}_{X^*}(\mathbf{x}_K)\|^2) \leq O\left(K^{-a} + L_f^2 L_g K^{-4a+4b} + L_f^2 K^{-b}\right). \tag{8}$$

*If $L_g \neq 0$ and $L_f \neq 0$, choose $a = 1$ and $b = 4/5$, yielding*

$$\mathbb{E}(\|\mathbf{x}_K - \mathcal{P}_{X^*}(\mathbf{x}_K)\|^2) \leq O(K^{-4/5}). \tag{9}$$

*If $L_g = 0$ or $L_f = 0$, choose $a = 1$ and $b = 1$, yielding*

$$\mathbb{E}(\|\mathbf{x}_K - \mathcal{P}_{X^*}(\mathbf{x}_K)\|^2) \leq O(K^{-1}). \tag{10}$$

Let us discuss the results of Theorem 2. In the general case where $L_f \neq 0$ and $L_g \neq 0$, the convergence rate in (9) is consistent with the result of Wang et al. [2016]. Now consider the special case where $L_g = 0$, i.e., the inner mapping is linear. This result finds an immediate application to Bellman error minimization problem (2) which arises from reinforcement learning problem in (and with $\ell_1$ norm regularization). The proposed ASC-PG algorithm is able to achieve the optimal rate $O(1/K)$ without any extra assumption on the outer function $f_v$. To the best of our knowledge, this is the best (also optimal) sample-error complexity for on-policy reinforcement learning.

**Remarks** Theorems 1 and 2 give important implications about the special cases where $L_f = 0$ or $L_g = 0$. In these cases, we argue that our convergence rate (10) is "*optimal*" with respect to the sample size $K$. To see this, it is worth pointing out the $O(1/K)$ rate of convergence is optimal for strongly convex expectation minimization problem. Because the expectation minimization problem is a special case of the problem (1), the $O(1/K)$ convergence rate must be optimal for the stochastic composition problem too.

- Consider the case where $L_f = 0$, which means that the outer function $f_v(\cdot)$ is linear with probability 1. Then the stochastic composition problem (1) reduces to an expectation minimization problem since $(\mathbb{E}_v f_v \circ \mathbb{E}_w g_w)(\mathbf{x}) = \mathbb{E}_v(f_v(\mathbb{E}_w g_w(\mathbf{x}))) = \mathbb{E}_v \mathbb{E}_w(f_v \circ g_w)(\mathbf{x})$. Therefore, it makes a perfect sense to obtain the optimal convergence rate.
- Consider the case where $L_g = 0$, which means that the inner function $g(\cdot)$ is a linear mapping. The result is quite surprising. Note that even $g(\cdot)$ is a linear mapping, it does not reduce problem (1) to an expectation minimization problem. However, the ASC-PG still achieves the optimal convergence rate. This suggests that, when inner linearity holds, the stochastic composition problem (1) is not fundamentally more difficult than the expectation minimization problem.

The convergence rate results unveiled in Theorems 1 and 2 are the best known results for the composition problem. We believe that they provide important new result which provides insights into the complexity of the stochastic composition problem.

## 4 Application to Reinforcement Learning

In this section, we apply the proposed ASC-PG algorithm to conduct policy value evaluation in reinforcement learning through attacking Bellman equations. Suppose that there are in total $S$ states. Let the policy of interest be $\pi$. Denote the value function of states by $V^\pi \in \Re^S$, where $V^\pi(s)$ denotes the value of being at state $s$ under policy $\pi$. The Bellman equation of the problem is

$$V^\pi(s_1) = \mathbb{E}_\pi\{r_{s_1,s_2} + \gamma \cdot V^\pi(s_2)|s_1\} \text{ for all } s_1, s_2 \in \{1, ..., S\},$$

where $r_{s_1,s_2}$ denotes the reward of moving from state $s_1$ to $s_2$, and the expectation is taken over all possible future state $s_2$ conditioned on current state $s_1$ and the policy $\pi$. We have that the solution $V^* \in \Re^S$ to the above equation satisfies that $V^* = V^\pi$. Here a moderately large $S$ will make solving the Bellman equation directly impractical. To resolve the curse of dimensionality, in many practical applications, we approximate the value of each state by some linear map of its feature $\phi_s \in \Re^m$, where $d < S$ to reduce the dimension. In particular, we assume that $V^\pi(s) \approx \phi_s^T \mathbf{w}^*$ for some $\mathbf{w}^* \in \Re^m$.

To compute $\mathbf{w}^*$, we formulate the problem as a Bellman residual minimization problem that

$$\min_{\mathbf{w}} \sum_{s=1}^{S} (\phi_s^T \mathbf{w} - q_{\pi,s'}(\mathbf{w}))^2,$$

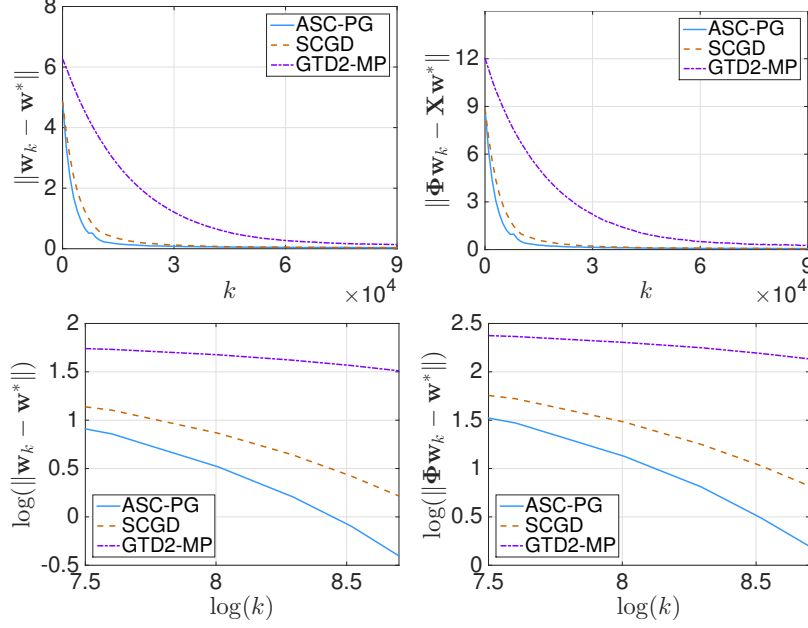

Figure 1: Empirical convergence rate of the ASC-PG algorithm and the GTD2-MP algorithm under Experiment 1 averaged over 100 runs, where $\mathbf{w}_k$ denotes the solution at the $k$-th iteration.

where $q_{\pi,s'}(\mathbf{w}) = \mathbb{E}_\pi\{r_{s,s'} + \gamma \cdot \boldsymbol{\phi}_{s'}\mathbf{w}\} = \sum_{s'} P^\pi_{ss'}(\{r_{s,s'} + \gamma \cdot \boldsymbol{\phi}_{s'}\mathbf{w}\}); \gamma < 1$ is a discount factor, and $r_{s,s'}$ is the random reward of transition from state $s$ to state $s'$. It is clearly seen that the proposed ASC-PG algorithm could be directly applied to solve this problem where we take

$$g(\mathbf{w}) = (\boldsymbol{\phi}_1^T\mathbf{w}, q_{\pi,1}(\mathbf{w}), ..., \boldsymbol{\phi}_S^T\mathbf{w}, q_{\pi,S}(\mathbf{w})) \in \Re^{2S},$$

$$f\Big((\boldsymbol{\phi}_1^T\mathbf{w}, q_{\pi,1}(\mathbf{w}), ..., \boldsymbol{\phi}_S^T\mathbf{w}, q_{\pi,S}(\mathbf{w}))\Big) = \sum_{s=1}^{S}(\boldsymbol{\phi}_s\mathbf{w} - q_{\pi,s'}(\mathbf{w}))^2 \in \Re.$$

We point out that the $g(\cdot)$ function here is a linear map. By our theoretical analysis, we expect to achieve a faster $O(1/k)$ rate, which is justified empirically in our later simulation study.

We consider three experiments, where in the first two experiments, we compare our proposed accelerated ASC-PG algorithm with SCGD algorithm [Wang et al., 2016] and the recently proposed GTD2-MP algorithm [Liu et al., 2015]. Also, in the first two experiments, we do not add any regularization term, i.e. $R(\cdot) = 0$. In the third experiment, we add an $\ell_1$-penalization term $\lambda\|\mathbf{w}\|_1$. In all cases, we choose the step sizes via comparison studies as in Dann et al. [2014]:

- Experiment 1: We use the Baird's example [Baird et al., 1995], which is a well-known example to test the off-policy convergent algorithms. This example contains $S = 6$ states, and two actions at each state. We refer the readers to Baird et al. [1995] for more detailed information of the example.
- Experiment 2: We generate a Markov decision problem (MDP) using similar setup as in White and White [2016]. In each instance, we randomly generate an MDP which contains $S = 100$ states, and three actions at each state. The dimension of the Given one state and one action, the agent can move to one of four next possible states. In our simulation, we generate the transition probabilities for each MDP instance uniformly from $[0, 1]$ and normalize the sum of transitions to one, and we generate the reward for each transition also uniformly in $[0, 1]$.
- Experiment 3: We generate the data same as Experiment 2 except that we have a larger $d = 100$ dimensional feature space, where only the first 4 components of $\mathbf{w}^*$ are non-zeros. We add an $\ell_1$-regularization term, $\lambda\|\mathbf{w}\|_1$, to the objective function.

Denote by $\mathbf{w}_k$ the solution at the $k$-th iteration. For the first two experiments, we report the empirical convergence performance $\|\mathbf{w}_k - \mathbf{w}^*\|$ and $\|\boldsymbol{\Phi}\mathbf{w}_k - \boldsymbol{\Phi}\mathbf{w}^*\|$, where $\boldsymbol{\Phi} = (\boldsymbol{\phi}_1, ..., \boldsymbol{\phi}_S)^T \in \Re^{S \times d}$ and $\boldsymbol{\Phi}\mathbf{w}^* = V$, and all $\mathbf{w}_k$'s are averaged over 100 runs, in the first two subfigures of Figures 1 and 2. It is seen that the ASC-PG algorithm achieves the fastest convergence rate empirically in both experiments. To further evaluate our theoretical results, we plot $\log(t)$ vs. $\log(\|\mathbf{w}_k - \mathbf{w}^*\|)$ (or $\log(\|\boldsymbol{\Phi}\mathbf{w}_k - \boldsymbol{\Phi}^*\|)$) averaged over 100 runs for the first two experiments in the second two subfigures of Figures 1 and

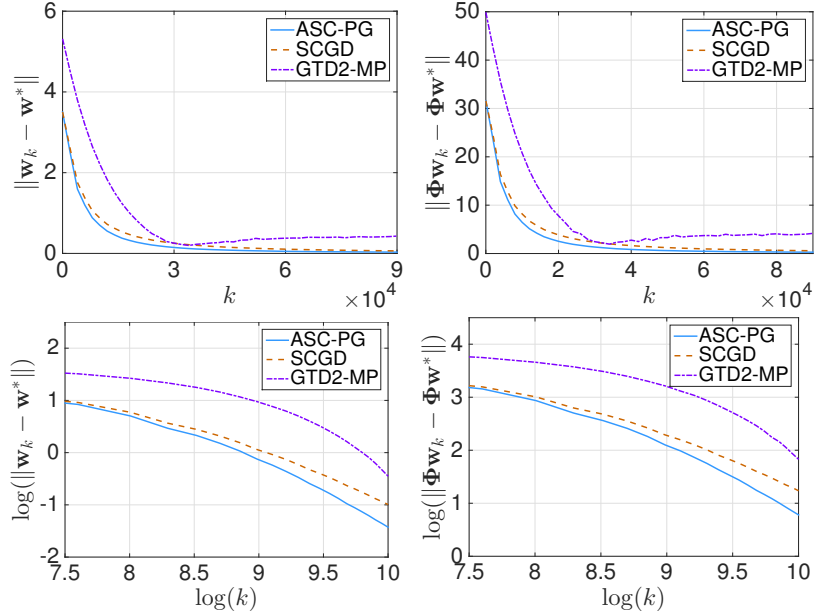

Figure 2: Empirical convergence rate of the ASC-PG algorithm and the GTD2-MP algorithm under Experiment 2 averaged over 100 runs, where $\mathbf{w}_k$ denotes the solution at the $k$-th iteration.

2. The empirical results further support our theoretical analysis that $\|\mathbf{w}_k - \mathbf{w}^*\|^2 = O(1/k)$ for the ASC-PG algorithm when $g(\cdot)$ is a linear mapping.

For Experiment 3, as the optimal solution is unknown, we run the ASC-PG algorithm for one million iterations and take the corresponding solution as the optimal solution $\hat{\mathbf{w}}^*$, and we report $\|\mathbf{w}_k - \hat{\mathbf{w}}^*\|$ and $\|\mathbf{\Phi}\mathbf{w}_k - \mathbf{\Phi}\hat{\mathbf{w}}^*\|$ averaged over 100 runs in Figure 3. It is seen the the ASC-PG algorithm achieves fast empirical convergence rate.

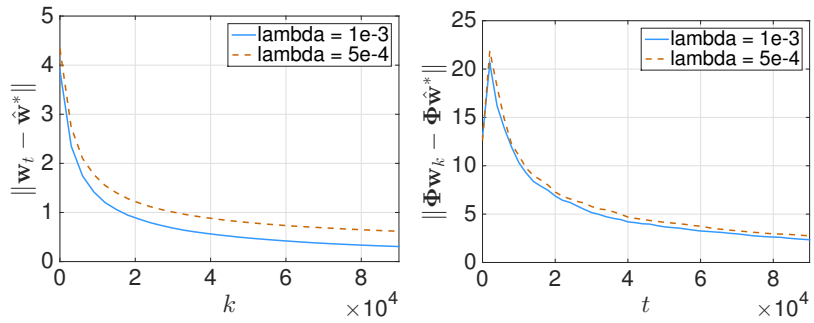

Figure 3: Empirical convergence rate of the ASC-PG algorithm with the $\ell_1$-regularization term $\lambda\|\mathbf{w}\|_1$ under Experiment 3 averaged over 100 runs, where $\mathbf{w}_k$ denotes the solution at the $t$-th iteration.

## 5   Conclusion

We develop a proximal gradient method for the penalized stochastic composition problem. The algorithm updates by interacting with a stochastic first-order oracle. Convergence rates are established under a variety of assumptions, which provide new rate benchmarks. Application of the ASC-PG to reinforcement learning leads to a new on-policy learning algorithm, which achieves faster convergence than the best known algorithms. For future research, it remains open whether or under what circumstances the current $O(K^{-4/9})$ can be further improved. Another direction is to customize and adapt the algorithm and analysis to more specific problems arising from reinforcement learning and risk-averse optimization, in order to fully exploit the potential of the proposed method.

## Acknowledgments

This project is in part supported by NSF grants CNS-1548078 and DMS-10009141.

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
