[Supplementary Material]

# Supplemental Materials

**Lemma 3.** *Under Assumption 3, two subsequent iterates in Algorithm 1 satisfy*

$$\|\mathbf{x}_k - \mathbf{x}_{k+1}\|^2 \leq \Theta(\alpha_k^2).$$

*Proof.* From the definition of the proximal operation, we have

$$
\begin{aligned}
\mathbf{x}_{k+1} &= \operatorname{prox}_{\alpha_k R(\cdot)}(\mathbf{x}_k - \alpha_k \nabla g_{w_k}^\top(\mathbf{x}_k) \nabla f_{v_k}(\mathbf{y}_k)) \\
&= \arg\min_x \frac{1}{2}\|\mathbf{x} - \mathbf{x}_k + \alpha_k \nabla g_{w_k}^\top(\mathbf{x}_k) \nabla f_{v_k}(\mathbf{y}_k)\|^2 + \alpha_k R(\mathbf{x}).
\end{aligned}
$$

The optimality condition suggests the following equality:

$$\mathbf{x}_{k+1} - \mathbf{x}_k = -\alpha_k(\nabla g_{w_k}^\top(\mathbf{x}_k)\nabla f_{v_k}(\mathbf{y}_k) + \mathbf{s}_{k+1}) \tag{11}$$

where $\mathbf{s}_{k+1} \in \partial R(\mathbf{x}_{k+1})$ is some vector in the sub-differential set of $R(\cdot)$ at $\mathbf{x}_{k+1}$. Then apply the boundedness condition in Assumption 3 to yield

$$
\begin{aligned}
\|\mathbf{x}_{k+1} - \mathbf{x}_k\| \quad &= \quad \alpha_k\|(\nabla g_{w_k}^\top(\mathbf{x}_k)\nabla f_{v_k}(\mathbf{y}_k) + \mathbf{s}_{k+1})\| \\
&\leq \quad \alpha_k(\|\nabla g_{w_k}^\top(\mathbf{x}_k)\nabla f_{v_k}(\mathbf{y}_k)\| + \|\mathbf{s}_{k+1}\|) \\
&\leq \quad \alpha_k(\|\nabla g_{w_k}^\top(\mathbf{x}_k)\|\|\nabla f_{v_k}(\mathbf{y}_k)\| + \|\mathbf{s}_{k+1}\|) \\
\text{(Assumption 3)} \quad &\leq \quad \Theta(1)\alpha_k,
\end{aligned}
$$

which implies the claim. $\qquad\square$

**Lemma 4.** *Under Assumptions 3 and 4, we have*

$$\|\nabla g_w^\top(\mathbf{x})\nabla_v f(g(\mathbf{x})) - \nabla g_w^\top(\mathbf{x})\nabla_v f(\mathbf{y}))\| \quad \leq \quad \Theta(L_f\|\mathbf{y} - g(\mathbf{x})\|).$$

*Proof.* We have

$$
\begin{aligned}
\|\nabla g_w^\top(\mathbf{x})\nabla f_v(g(\mathbf{x})) - \nabla g_w^\top(\mathbf{x})\nabla f_v(\mathbf{y}))\| \quad &\leq \quad \|\nabla g_w^\top(\mathbf{x})\|\|\nabla f_v(g(\mathbf{x})) - \nabla f_v(\mathbf{y})\| \\
\text{(Assumption 3)} \quad &\leq \quad \Theta(1)\|\nabla f_v(g(\mathbf{x})) - \nabla f_v(\mathbf{y})\| \\
\text{(Assumption 4)} \quad &\leq \quad \Theta(L_f)\|\mathbf{y} - g(\mathbf{x})\|.
\end{aligned}
$$

It completes the proof. $\qquad\square$

**Lemma 5.** *Given a positive sequence $\{w_k\}_{k=1}^\infty$ satisfying*

$$w_{k+1} \leq (1 - \beta_k + C_1\beta_k^2)w_k + C_2 k^{-a} \tag{12}$$

*where $C_1 \geq 0$, $C_2 \geq 0$, and $a \geq 0$. Choosing $\beta_k$ to be $\beta_k = C_3 k^{-b}$ where $b \in (0,1]$ and $C_3 > 2$, the sequence can be bounded by*

$$w_k \leq C k^{-c}$$

*where $C$ and $c$ are defined as*

$$C := \max_{k \leq (C_1 C_3^2)^{1/b}+1} w_k k^c + \frac{C_2}{C_3 - 2} \quad \text{and} \quad c := a - b.$$

*In other words, we have*

$$w_k \leq \Theta(k^{-a+b}).$$

*Proof.* We prove it by induction. First it is easy to verify that the claim holds for $k \leq (C_1 C_3^2)^{1/b}$ from the definition for $C$. Next we prove from "$k$" to "$k+1$", that is, given $w_k \leq Ck^{-c}$ for $k > (C_1 C_3^2)^{1/b}$, we need to prove $w_{k+1} \leq C(k+1)^{-c}$.

$$
\begin{aligned}
w_{k+1} \quad &\leq \quad (1 - \beta_k + C_1\beta_k^2)w_k + C_2 k^{-a} \\
&\leq \quad (1 - C_3 k^{-b} + C_1 C_3^2 k^{-2b})Ck^{-c} + C_2 k^{-a}
\end{aligned}
$$

$$= \quad Ck^{-c} - CC_3k^{-b-c} + CC_1C_3^2k^{-2b-c} + C_2k^{-a}. \tag{13}$$

To prove that (13) is bounded by $C(k+1)^{-c}$, it suffices to show that

$$\Delta := (k+1)^{-c} - k^{-c} + C_3k^{-b-c} - C_1C_3^2k^{-2b-c} > 0 \quad \text{and} \quad C \geq \frac{C_2k^{-a}}{\Delta}.$$

From the convexity of function $h(t) = t^{-c}$, we have the inequality $(k+1)^{-c} - k^{-c} \geq (-c)k^{-c-1}$. Therefore we obtain

$$\Delta \quad \underset{(b\leq 1,\, k>(C_1C_3^2)^{1/b})}{\geq} \quad -ck^{-c-1} + C_3k^{-b-c} - C_1C_3^2k^{-2b-c}$$

$$\underset{(C_3>2)}{\geq} \quad (C_3 - 2)(k^{-b-c})$$

$$> \quad 0.$$

To verify the second one, we have

$$\frac{C_2k^{-a}}{\Delta} \leq \frac{C_2}{C_3-2}k^{-a+b+c} \overset{(c=a+b)}{=} \frac{C_2}{C_3-2} \leq C$$

where the last inequality is due to the definition of $C$. It completes the proof. $\square$

**Lemma 6.** *Choose $\beta_k$ to be $\beta_k = C_b k^{-b}$ where $C_b > 2$, $b \in (0,1]$, and $\alpha_k = C_a k^{-a}$. Under Assumptions 1 and 2, we have*

$$\mathbb{E}(\|\mathbf{y}_k - g(\mathbf{x}_k)\|^2) \leq L_g \Theta(k^{-4a+4b}) + \Theta(k^{-b}). \tag{14}$$

*Proof.* Denote by $m_{k+1}$

$$m_{k+1} := \sum_{t=0}^{k} \xi_t^{(k)} \|\mathbf{x}_{k+1} - \mathbf{z}_{t+1}\|^2$$

and $n_{k+1}$

$$n_{k+1} := \left\| \sum_{t=0}^{k} \xi_t^{(k)} (g_{w_{t+1}}(\mathbf{z}_{t+1}) - g(\mathbf{z}_{t+1})) \right\|$$

for short.

From Lemma 10 in [Wang et al., 2016], we have

$$\|\mathbf{y}_k - g(\mathbf{x}_k)\|^2 \leq \left( \frac{L_g}{2} m_k + n_k \right)^2 \leq L_g m_k^2 + 2n_k^2. \tag{15}$$

From Lemma 11 in [Wang et al., 2016], $m_{k+1}$ can be bounded by

$$m_{k+1} \quad \leq \quad (1 - \beta_k)m_k + \beta_k q_k + \frac{2}{\beta_k}\|\mathbf{x}_k - \mathbf{x}_{k+1}\|^2 \tag{16}$$

where $q_k$ is bounded by

$$q_{k+1} \quad \leq \quad (1 - \beta_k)q_k + \frac{4}{\beta_k}\|\mathbf{x}_{k+1} - \mathbf{x}_k\|^2$$

$$\overset{\text{(Lemma 3)}}{\leq} \quad (1 - \beta_k)q_k + \frac{\Theta(1)\alpha_k^2}{\beta_k}$$

$$\leq \quad (1 - \beta_k)q_k + \Theta(k^{-2a+b}).$$

Use Lemma 5 and obtain the following decay rate

$$q_k \leq \Theta(k^{-2a+2b}).$$

Together with (16), we have

$$m_{k+1} \quad \leq \quad (1 - \beta_k)m_k + \beta_k q_k + \frac{2}{\beta_k}\|\mathbf{x}_k - \mathbf{x}_{k+1}\|^2$$

$$\leq \quad (1-\beta_k)m_k + \Theta(k^{-2a+b}) + \Theta(k^{-2a+b})$$
$$\leq \quad (1-\beta_k)m_k + \Theta(k^{-2a+b}),$$

which leads to

$$m_k \leq \Theta(k^{-2a+2b}) \quad \text{and} \quad m_k^2 \leq \Theta(k^{-4a+4b}). \tag{17}$$

by using Lemma 5 again. Then we estimate the upper bound for $\mathbb{E}(n_k^2)$. From Lemma 11 in [Wang et al., 2016], we know $\mathbb{E}(n_k^2)$ is bounded by

$$\mathbb{E}(n_{k+1}^2) \leq (1-\beta_k)^2 \mathbb{E}(\|n_k\|^2) + \beta_k^2 \sigma_g^2 = (1-2\beta_k+\beta_k^2)\mathbb{E}(\|n_k\|^2) + \beta_k^2 \sigma_g^2.$$

By using Lemma 5 again, we have

$$\mathbb{E}(n_k^2) \leq \Theta(k^{-b}). \tag{18}$$

Now we are ready to estimate the upper bound of $\|\mathbf{y}_{k+1} - g(\mathbf{x}_{k+1})\|^2$ by following (15)

$$\mathbb{E}(\|\mathbf{y}_k - g(\mathbf{x}_k)\|^2) \quad \leq \quad L_g \mathbb{E}(m_k^2) + 2\mathbb{E}(n_k^2)$$
$$\overset{(17)+(18)}{\leq} \quad L_g \Theta(k^{-4a+4b}) + \Theta(k^{-b}).$$

It completes the proof. $\qquad\square$

**Proof to Theorem 1**

*Proof.* From the Lipschitzian condition in Assumption 4, we have

$$F(\mathbf{x}_{k+1}) - F(\mathbf{x}_k)$$
$$\leq \quad \langle \nabla F(\mathbf{x}_k),\ \mathbf{x}_{k+1} - \mathbf{x}_k \rangle + \frac{L_F}{2}\|\mathbf{x}_{k+1} - \mathbf{x}_k\|^2$$
$$\overset{\text{(Lemma 5)}}{\leq} \quad -\alpha_k \langle \nabla F(\mathbf{x}_k),\ \nabla g_{w_k}^\top(\mathbf{x}_k)\nabla f_{v_k}(\mathbf{y}_k)\rangle + \Theta(\alpha_k^2)$$
$$= \quad -\alpha_k\|\nabla F(\mathbf{x}_k)\|^2 + \alpha_k \underbrace{\langle \nabla F(\mathbf{x}_k),\ \nabla F(\mathbf{x}_k) - \nabla g_{w_k}^\top(\mathbf{x}_k)\nabla f_{v_k}(\mathbf{y}_k)\rangle}_{=:T}$$
$$+ \Theta(\alpha_k^2) \tag{19}$$

Next we estimate the upper bound for $\mathbb{E}(T)$:

$$\mathbb{E}(T) \quad = \quad \mathbb{E}(\langle \nabla F(\mathbf{x}_k),\ \nabla F(\mathbf{x}_k) - \nabla g_{w_k}^\top(\mathbf{x}_k)\nabla f_{v_k}(g(\mathbf{x}_k))\rangle)$$
$$+ \mathbb{E}(\langle \nabla F(\mathbf{x}_k),\ \nabla g_{w_k}^\top(\mathbf{x}_k)\nabla f_{v_k}(g(\mathbf{x}_k)) - \nabla g_{w_k}^\top(\mathbf{x}_k)\nabla f_{v_k}(\mathbf{y}_k)\rangle)$$
$$\overset{\text{(Assumption 1)}}{=} \quad \mathbb{E}(\langle \nabla F(\mathbf{x}_k),\ \nabla g_{w_k}^\top(\mathbf{x}_k)\nabla f_{v_k}(g(\mathbf{x}_k)) - \nabla g_{w_k}^\top(\mathbf{x}_k)\nabla f_{v_k}(\mathbf{y}_k)\rangle)$$
$$\leq \quad \frac{1}{2}\mathbb{E}(\|\nabla F(\mathbf{x}_k)\|^2) + \frac{1}{2}\mathbb{E}(\|\nabla g_{w_k}^\top(\mathbf{x}_k)\nabla f_{v_k}(g(\mathbf{x}_k)) - \nabla g_{w_k}^\top(\mathbf{x}_k)\nabla f_{v_k}(\mathbf{y}_k)\|^2)$$
$$\overset{\text{(Lemma 4)}}{\leq} \quad \frac{1}{2}\mathbb{E}(\|\nabla F(\mathbf{x}_k)\|^2) + \Theta(L_f^2)\mathbb{E}(\|\mathbf{y}_k - g(\mathbf{x}_k)\|^2).$$

Take expectation on both sides of (19) and substitute $\mathbb{E}(T)$ by its upper bound:

$$\frac{\alpha_k}{2}\|\nabla F(\mathbf{x}_k)\|^2$$
$$\leq \quad \mathbb{E}(F(\mathbf{x}_k)) - \mathbb{E}(F(\mathbf{x}_{k+1})) + \Theta(L_f^2\alpha_k)\mathbb{E}(\|\mathbf{y}_k - g(\mathbf{x}_k)\|^2) + \Theta(\alpha_k^2)$$
$$\overset{\text{(Lemma 6)}}{\leq} \quad \mathbb{E}(F(\mathbf{x}_k)) - \mathbb{E}(F(\mathbf{x}_{k+1})) + L_g\Theta(L_f^2\alpha_k)\Theta(k^{-4a+4b}) + \Theta(L_f^2\alpha_k k^{-b}) + \Theta(\alpha_k^2)$$
$$\leq \quad \mathbb{E}(F(\mathbf{x}_k)) - \mathbb{E}(F(\mathbf{x}_{k+1})) + L_f^2 L_g\Theta(k^{-5a+4b}) + L_f^2\Theta(k^{-a-b}) + \Theta(k^{-2a})$$

which suggests that

$$\mathbb{E}(\|\nabla F(\mathbf{x}_k)\|^2)$$
$$\leq \quad 2\alpha_k^{-1}\mathbb{E}(F(\mathbf{x}_k)) - 2\alpha_k^{-1}\mathbb{E}(F(\mathbf{x}_{k+1})) + L_f^2 L_g\Theta(k^{-4a+4b}) + L_f^2\Theta(k^{-b}) + \Theta(k^{-a})$$

$$\leq \quad 2k^a\mathbb{E}(F(\mathbf{x}_k)) - 2k^a\mathbb{E}(F(\mathbf{x}_{k+1})) + L_f^2 L_g\Theta(k^{-4a+4b}) + L_f^2\Theta(k^{-b}) + \Theta(k^{-a}). \quad (20)$$

Summarize Eq. (20) from $k = 1$ to $K$ and obtain

$$
\begin{aligned}
\frac{\sum_{k=1}^K \mathbb{E}(\|\nabla F(\mathbf{x}_k)\|^2)}{K} \quad &\leq \quad 2K^{-1}\alpha_1^{-1}F(\mathbf{x}_1) + K^{-1}\sum_{k=2}^K ((k+1)^a - k^a)\mathbb{E}(F(\mathbf{x}_k)) \\
&\quad + K^{-1}\sum_{k=1}^K L_f^2 L_g\Theta(k^{-4a+4b}) + K^{-1}L_f^2\sum_{k=1}^K \Theta(k^{-b}) + K^{-1}\sum_{k=1}^K \Theta(k^{-a}) \\
&\leq \quad 2K^{-1}F(\mathbf{x}_0) + K^{-1}\sum_{k=2}^K ak^{a-1}\mathbb{E}(F(\mathbf{x}_k)) \\
&\quad + K^{-1}\sum_{k=1}^K L_f^2 L_g\Theta(k^{-4a+4b}) + K^{-1}L_f^2\sum_{k=1}^K \Theta(k^{-b}) + K^{-1}\sum_{k=1}^K \Theta(k^{-a}) \\
&\leq \quad O(K^{a-1} + L_f^2 L_g K^{4b-4a}\mathbf{I}_{4a-4b=1}^{\log K} + L_f^2 K^{-b} + K^{-a}),
\end{aligned}
$$

where the second inequality uses the fact that $h(t) = t^a$ is a concave function suggesting $(k+1)^a \leq k^a + ak^{a-1}$, and the last inequality uses the condition $\mathbb{E}(F(\mathbf{x}_k)) \leq \Theta(1)$.

The optimal $a^* = 5/9$ and the optimal $b^* = 4/9$, which leads to the convergence rate $O(K^{-4/9})$. □

**Lemma 7.** *Assume that both $F(\mathbf{x})$ and $R(\mathbf{x})$ are convex. Under Assumptions 1, 2, 3, and 4, the iterates generated by Algorithm 1 satisfies for any sequence of positive scalars $\{\phi_k\}$:*

$$
\begin{aligned}
2\alpha_k(\mathbb{E}(H(\mathbf{x}_{k+1})) - H^*) + \mathbb{E}(\|\mathbf{x}_{k+1} - \mathcal{P}_{X^*}(\mathbf{x}_{k+1})\|^2) & \quad\quad\quad (P_k) \\
\leq (1+\phi_k)\mathbb{E}(\|\mathbf{x}_k - \mathcal{P}_{X^*}(\mathbf{x}_k)\|^2) + \Theta(\alpha_k^3) + \Theta(L_f^2\alpha_k^2/\phi_k)\mathbb{E}(\|\mathbf{y}_k - g(\mathbf{x}_k)\|^2) + \Theta(\alpha_k^2).
\end{aligned}
$$

*Proof.* Following the line of the proof to Lemma 3, we have

$$\mathbf{x}_{k+1} - \mathbf{x}_k = -\alpha_k(\nabla g_{w_k}^\top(\mathbf{x}_k)\nabla f_{v_k}(\mathbf{y}_k) + \mathbf{s}_{k+1}) \quad (21)$$

where $\mathbf{s}_{k+1} \in \partial R(\mathbf{x}_{k+1})$ is some vector in the sub-differential set of $R(\cdot)$ at $\mathbf{x}_{k+1}$. Then we consider $\|\mathbf{x}_{k+1} - \mathcal{P}_{X^*}(\mathbf{x}_{k+1})\|^2$:

$$
\begin{aligned}
&\quad \|\mathbf{x}_{k+1} - \mathcal{P}_{X^*}(\mathbf{x}_{k+1})\|^2 \\
&\leq \quad \|\mathbf{x}_{k+1} - \mathbf{x}_k + \mathbf{x}_k - \mathcal{P}_{X^*}(\mathbf{x}_k)\|^2 \\
&= \quad \|\mathbf{x}_k - \mathcal{P}_{X^*}(\mathbf{x}_k)\|^2 - \|\mathbf{x}_{k+1} - \mathbf{x}_k\|^2 + 2\langle\mathbf{x}_{k+1} - \mathbf{x}_k, \mathbf{x}_{k+1} - \mathcal{P}_{X^*}(\mathbf{x}_k)\rangle \\
&\overset{(21)}{=} \quad \|\mathbf{x}_k - \mathcal{P}_{X^*}(\mathbf{x}_k)\|^2 - \|\mathbf{x}_{k+1} - \mathbf{x}_k\|^2 - 2\alpha_k\langle\nabla g_{w_k}^\top(\mathbf{x}_k)\nabla f_{v_k}(\mathbf{y}_k) + \mathbf{s}_{k+1}, \mathbf{x}_{k+1} - \mathcal{P}_{X^*}(\mathbf{x}_k)\rangle \\
&= \quad \|\mathbf{x}_k - \mathcal{P}_{X^*}(\mathbf{x}_k)\|^2 - \|\mathbf{x}_{k+1} - \mathbf{x}_k\|^2 + 2\alpha_k\langle\nabla g_{w_k}^\top(\mathbf{x}_k)\nabla f_{v_k}(\mathbf{y}_k), \mathcal{P}_{X^*}(\mathbf{x}_k) - \mathbf{x}_{k+1}\rangle \\
&\quad + 2\alpha_k\langle\mathbf{s}_{k+1}, \mathcal{P}_{X^*}(\mathbf{x}_k) - \mathbf{x}_{k+1}\rangle \\
&\leq \quad \|\mathbf{x}_k - \mathcal{P}_{X^*}(\mathbf{x}_k)\|^2 - \|\mathbf{x}_{k+1} - \mathbf{x}_k\|^2 + 2\alpha_k\langle\nabla g_{w_k}^\top(\mathbf{x}_k)\nabla f_{v_k}(\mathbf{y}_k), \mathcal{P}_{X^*}(\mathbf{x}_k) - \mathbf{x}_{k+1}\rangle \\
&\quad + 2\alpha_k(R(\mathcal{P}_{X^*}(\mathbf{x}_k)) - R(\mathbf{x}_{k+1})) \quad \text{(due to the convexity of } R(\cdot)\text{)} \\
&\leq \quad \|\mathbf{x}_k - \mathcal{P}_{X^*}(\mathbf{x}_k)\|^2 - \|\mathbf{x}_{k+1} - \mathbf{x}_k\|^2 + 2\alpha_k\underbrace{\langle\nabla F(\mathbf{x}_k), \mathcal{P}_{X^*}(\mathbf{x}_k) - \mathbf{x}_{k+1}\rangle}_{T_1} \\
&\quad + 2\alpha_k\underbrace{\langle\nabla g_{w_k}^\top(\mathbf{x}_k)\nabla f_{v_k}(\mathbf{y}_k) - \nabla F(\mathbf{x}_k), \mathcal{P}_{X^*}(\mathbf{x}_k) - \mathbf{x}_{k+1}\rangle}_{T_2} \\
&\quad + 2\alpha_k(R(\mathcal{P}_{X^*}(\mathbf{x}_k)) - R(\mathbf{x}_{k+1})) \quad\quad\quad (22)
\end{aligned}
$$

where the second equality follows from $\|a+b\|^2 = \|b\|^2 - \|a\|^2 + 2\langle a, a+b\rangle$ with $a = \mathbf{x}_{k+1} - \mathbf{x}_k$ and $b = \mathbf{x}_k - \mathcal{P}_{X^*}(\mathbf{x}_k)$. We next estimate the upper bound for $T_1$ and $T_2$ respectively:

$$T_1 \quad = \quad \langle\nabla F(\mathbf{x}_k), \mathbf{x}_k - \mathbf{x}_{k+1}\rangle + \langle\nabla F(\mathbf{x}_k), -\mathbf{x}_k + \mathcal{P}_{X^*}(\mathbf{x}_k)\rangle$$

$$\leq \quad \underbrace{F(\mathbf{x}_k) - F(\mathbf{x}_{k+1}) + \frac{L_F}{2}\|\mathbf{x}_{k+1} - \mathbf{x}_k\|^2}_{\text{due to Assumption 4}} + \underbrace{F(\mathcal{P}_{X^*}(\mathbf{x}_k)) - F(\mathbf{x}_k)}_{\text{due to the convexity of } F(\cdot)}$$

$$= \quad F(\mathcal{P}_{X^*}(\mathbf{x}_k)) - F(\mathbf{x}_{k+1}) + \frac{L_F}{2}\|\mathbf{x}_{k+1} - \mathbf{x}_k\|^2$$

$$\leq \quad F(\mathcal{P}_{X^*}(\mathbf{x}_k)) - F(\mathbf{x}_{k+1}) + \Theta(\alpha_k^2),$$

where the last inequality uses Lemma 3.

$$
\begin{aligned}
T_2 &= \langle \nabla F(\mathbf{x}_k) - \nabla g_{w_k}^\top(\mathbf{x}_k)\nabla f_{v_k}(\mathbf{y}_k),\ \mathbf{x}_k - \mathcal{P}_{X^*}(\mathbf{x}_k)\rangle \\
&\quad + \langle \nabla F(\mathbf{x}_k) - \nabla g_{w_k}^\top(\mathbf{x}_k)\nabla f_{v_k}(\mathbf{y}_k),\ \mathbf{x}_{k+1} - \mathbf{x}_k\rangle \\
&\leq \underbrace{\langle \nabla F(\mathbf{x}_k) - \nabla g_{w_k}^\top(\mathbf{x}_k)\nabla f_{v_k}(g(\mathbf{x}_k)),\ \mathbf{x}_k - \mathcal{P}_{X^*}(\mathbf{x}_k)\rangle}_{T_{2,1}} \\
&\quad + \underbrace{\langle \nabla g_{w_k}^\top(\mathbf{x}_k)\nabla f_{v_k}(g(\mathbf{x}_k)) - \nabla g_{w_k}^\top(\mathbf{x}_k)\nabla f_{v_k}(\mathbf{y}_k),\ \mathbf{x}_k - \mathcal{P}_{X^*}(\mathbf{x}_k)\rangle}_{T_{2,2}} \\
&\quad + \frac{\alpha_k}{2}\underbrace{\|\nabla F(\mathbf{x}_k) - \nabla g_{w_k}^\top(\mathbf{x}_k)\nabla f_{v_k}(\mathbf{y}_k)\|^2}_{T_{2,3}} + \frac{1}{2\alpha_k}\|\mathbf{x}_k - \mathbf{x}_{k+1}\|^2
\end{aligned}
$$

where the last line is due to the inequality $\langle a, b\rangle \leq \frac{1}{2\alpha_k}\|a\|^2 + \frac{\alpha_k}{2}\|b\|^2$. For $T_{2,1}$, we have $\mathbb{E}(T_{2,1}) = 0$ due to Assumption 1. For $T_{2,2}$, we have

$$
\begin{aligned}
T_{2,2} \quad &\leq \quad \frac{\alpha_k}{2\phi_k}\|\nabla g_{w_k}^\top(\mathbf{x}_k)\nabla f_{v_k}(g(\mathbf{x}_k)) - \nabla g_{w_k}^\top(\mathbf{x}_k)\nabla f_{v_k}(\mathbf{y}_k)\|^2 + \frac{\phi_k}{2\alpha_k}\|\mathbf{x}_k - \mathcal{P}_{X^*}(\mathbf{x}_k)\|^2 \\
&\overset{\text{(Lemma 4)}}{\leq} \quad \Theta\left(L_f^2\frac{\alpha_k}{\phi_k}\right)\|\mathbf{y}_k - g(\mathbf{x}_k)\|^2 + \frac{\phi_k}{2\alpha_k}\|\mathbf{x}_k - \mathbf{x}_{k+1}\|^2.
\end{aligned}
$$

$T_{2,3}$ can be bounded by a constant

$$T_{2,3} \leq 2\|\nabla F(\mathbf{x}_k)\|^2 + 2\|\nabla g_{w_k}^\top \nabla f_{v_k}(\mathbf{y}_k)\|^2 \overset{\text{(Assumption 3)}}{\leq} \Theta(1).$$

Take expectation on $T_2$ and put all pieces into it:

$$\mathbb{E}(T_2) \quad \leq \quad \Theta\left(L_f^2\frac{\alpha_k}{\phi_k}\right)\|\mathbf{y}_k - g(\mathbf{x}_k)\|^2 + \frac{1}{2\alpha_k}(\phi_k\|\mathbf{x}_k - \mathcal{P}_{X^*}(\mathbf{x}_k)\|^2 + \|\mathbf{x}_k - \mathbf{x}_{k+1}\|^2) + \Theta(\alpha_k).$$

Taking expectation on both sides of (22) and plugging the upper bounds of $T_1$ and $T_2$ into it, we obtain

$$
\begin{aligned}
&2\alpha_k(\mathbb{E}(H(\mathbf{x}_{k+1})) - H^*) + \mathbb{E}(\|\mathbf{x}_{k+1} - \mathcal{P}_{X^*}(\mathbf{x}_{k+1})\|^2) \\
&\quad \leq (1 + \phi_k)\mathbb{E}(\|\mathbf{x}_k - \mathcal{P}_{X^*}(\mathbf{x}_k)\|^2) + \Theta(\alpha_k^3) + \Theta(L_f^2\alpha_k^2/\phi_k)\mathbb{E}(\|\mathbf{y}_k - g(\mathbf{x}_k)\|^2) + \Theta(\alpha_k^2),
\end{aligned}
$$

which completes the proof. $\qquad\square$

**Proof to Theorem 2**

*Proof.* Apply the optimally strong convexity in (7) to Lemma 7, yielding

$$
\begin{aligned}
&(1 + 2\lambda\alpha_k)\mathbb{E}(\|\mathbf{x}_{k+1} - \mathcal{P}_{X^*}(\mathbf{x}_{k+1})\|^2) \\
&\quad \leq (1 + \phi_k)\mathbb{E}(\|\mathbf{x}_k - \mathcal{P}_{X^*}(\mathbf{x}_k)\|^2) + \Theta(\alpha_k^3) + \Theta(L_f^2\alpha_k^2/\phi_k)\mathbb{E}(\|\mathbf{y}_k - g(\mathbf{x}_k)\|^2) + \Theta(\alpha_k^2).
\end{aligned}
$$

It follows by dividing $1 + 2\lambda\alpha_k$ on both sides

$$
\begin{aligned}
&\mathbb{E}(\|\mathbf{x}_{k+1} - \mathcal{P}_{X^*}(\mathbf{x}_{k+1})\|^2) \\
&\quad \leq \frac{1 + \phi_k}{1 + 2\lambda\alpha_k}\mathbb{E}(\|\mathbf{x}_k - \mathcal{P}_{X^*}(\mathbf{x}_k)\|^2) + \Theta(\alpha_k^3) + \Theta(L_f^2\alpha_k^2/\phi_k)\mathbb{E}(\|\mathbf{y}_k - g(\mathbf{x}_k)\|^2) + \Theta(\alpha_k^2).
\end{aligned}
$$

Choosing $\phi_k = \lambda\alpha_k - 2\lambda^2\alpha_k^2 \geq 0.5\lambda\alpha_k$ yields

$$\mathbb{E}(\|\mathbf{x}_{k+1} - \mathcal{P}_{X^*}(\mathbf{x}_{k+1})\|^2)$$

$$\leq \quad (1-\lambda\alpha_k)\mathbb{E}(\|\mathbf{x}_k - \mathcal{P}_{X^*}(\mathbf{x}_k)\|^2) + \Theta(\alpha_k^2) + \frac{\Theta(L_f^2\alpha_k)}{\lambda}\mathbb{E}(\|g(\mathbf{x}_k) - \mathbf{y}_k\|^2)$$

$$\leq \quad (1-\lambda\alpha_k)\mathbb{E}(\|\mathbf{x}_k - \mathcal{P}_{X^*}(\mathbf{x}_k)\|^2) + \Theta(k^{-2a}) + \Theta(L_g L_f^2 k^{-5a+4b} + L_f^2 k^{-a-b}).$$

Apply Lemma 5 and substitute the subscript $k$ by $K$ to obtain the first claim in (8)

$$\mathbb{E}(\|\mathbf{x}_K - \mathcal{P}_{X^*}(\mathbf{x}_K)\|^2) \leq O\left(K^{-a} + L_f^2 L_g K^{-4a+4b} + L_f^2 K^{-b}\right).$$

The followed specification of $a$ and $b$ can easily verified. $\qquad\square$