[Reviews · NeurIPS 2016]

Reviewer 1

Summary

The paper has some apparent deficiency. Under Assumption 1,the SO will provide unbiased estimator of the gradient for the composition function and hence we can use existing algorithms to solve the problem of interest. The authors' account for existing works on expectation minimization problem appears to incomplete (see the end of Page 2) and miss a few important works in optimization and machine learning areas.

Qualitative Assessment

I believe the authors made unnecessarily strong assumptions. Under these assumptions, stronger convergence results can be obtained by using existing algorithms. The paper also missed quite a few critical references.

Confidence in this Review

2-Confident (read it all; understood it all reasonably well)


Reviewer 2

Summary

The paper extends the accelerated SCGD algorithm in [Wang et.al, 2016] with proximal setting to handle nonsmooth regularization, and present refined convergence results in some cases. Some experiment is conducted on reinforcement learning on small synthetic dataset.

Qualitative Assessment

First of all, the novelty and originality is a big concern. - Most elements are inherited from an early journal paper [Wang et.al, 2016], including the motivation example, the accelerated algorithm itself, results for strongly convex case, and application to dynamic programming. - The only novelty of the paper is to extend ASCG to proximal setting, which however, is quite trivial. Not to mention the fact that, the main Theorem 1 assumes no nonsmooth penalty term. So literally, this paper has not really done what they claimed - address nonsmooth regularization penalty without deteriorating the convergence rate. -The "optimally" strongly convexity condition is not substantially different from strongly convexity. Theorem 2 is more or less just a repetition of Theorem 7 in [Wang et.al, 2016], except with a slightly more general choice of learning rate. Overall, I don't feel the paper has a substantial contribution. Secondly, several statements are confusing and could be misleading. 1. The authors claim that the inner and outer random variables w, v can be dependent, but assumes unbiased stochastic oracles given in Assumption 1.1. This seems to be a contradiction, if (w,v) are dependent, then Assumption 1.1 might not be possible. So it feels like the algorithm is limited to independent (w,v). Due to this same reason, the proposed algorithm might not handle the motivating example, risk-averse learning, since the inner and outer random variables are the same. 2. Somehow the two motivating examples cannot be directly or explicitly reformulated to the form (1). I would suggest the authors to provide some details here on what are the exact functions f, g. Also, the reformulation (3) does not look quite right to me. 3. In theorem 1, the convergence criterion (taking average of gradient norms) is not common nor expressive enough to imply the rate. It would be more useful to see directly how the gradient of the last iterate converge to zero. Since the criterion used here is different from [Wang et.al, 2016], it is not very appropriate to compare the rates or claim any kind of improvement. Thirdly, the applicability of the proposed algorithm in reinforcement learning seems quite limited. - In order to translate the Bellman residual minimization problem into (1), an extra O(S) inner expectation terms needed to be introduced, which does not look very elegant. Also, this implies that the proposed algorithm can only handle finite state MDP with reasonably small number of states. As shown in their experiments, the largest instance has only 100 states. - For the interest of machine learning community, results on real and large-scale datasets should be provided. Lastly, here are some minor issues. 1. The authors claim that the objective function (3) is optimally strongly convex and later use it in experiments, so either proof or references should be provided. 2. In Experiment 1 and 2, there is no nonsmooth penalty, so the proposed ASC-PG algorithm is almost nothing different from the ASC-GD algorithm except with a smarter choice of learning rate. The authors should clarify this when making the comparison. In Experiment 3, it also makes sense to compare to the proximal extension of GTD2-MP algorithm. 3. The following reference, also provides a O(1/t) convergence rate for stochastic composition problems when inner function is linear. The authors might want to provides some discussion on the algorithmic differences. Dai, Bo, et al. "Scalable kernel methods via doubly stochastic gradients." Advances in Neural Information Processing Systems. 2014. 4. Line 139: function F(x) is not defined until in line 144. 5. Line 53: nonlinearity in distribution of v. Shouldn't it be w? 6. Line 228: ||w_k-w^*||=O(1/k) missing square?

Confidence in this Review

2-Confident (read it all; understood it all reasonably well)


Reviewer 3

Summary

This paper introduces and analyses an algorithm for the stochastic composition problem. It builds on the theory developed recently by Wang et al. The new algorithm is proved to have a smaller worst case complexity. The experiment section illustrates well the generality of the approach.

Qualitative Assessment

The paper is well motivated and the applications are numerous. Although most of the theory comes from Wang et al's paper, there are some new interesting results and a notable application to reinforcement learning. My major comment comes from Assumption 3. There is no asymptotics in this assumption. You should use Mf, Mg and MR for the Lipschitz constants of the functions rather than writing O(1) that means nothing in this context. Moreover, this new notation would be more consistent with Assumption 4 on the Lipschitz constants of the gradients. In fact, I abandoned checking all the proofs because of this notation that mixes limits on the Lipschitz constants and limits on the iteration number. Minor comments Line 19 vs Assumption 3: Infinite valued functions cannot have bounded subgradients. Hence, you should state from the beginning that R will be a real-valued function. Algorithm 1, step 4: Please write clearly that you query the oracle for g_{w_{k+1}}(z_{k+1}) before updating y in the extrapolation-smoothing step (for instance in a separate step). Line 136, Assumption 3: for all x is repeated twice Line 146: K^{1-a} -> K^{a-1} Line 201-202: The function f defined for this experiment does not have bounded gradients. Could you please justify why your analysis still applies

Confidence in this Review

2-Confident (read it all; understood it all reasonably well)


Reviewer 4

Summary

This paper proposed a new method for solving stochastic composition problem, which improves the convergence rate from k^{-2/7} to k^{-4/9}. This new method mainly follows from [Wang et al 2016], but with an acceleration technique. Besides, it can deal with non-smooth regularizer by proximal mapping. And its experimental performance surpasses existing methods.

Qualitative Assessment

This paper is well written and well motivated. The theoretical results are good and numerical experiments are convincing. Overall it is a good paper. My only concern is that some of the assumptions are not well justified. For example, Assumption 1 requires the samples gradients are unbiased. It is better to discuss what kind of functions satisfy this kind of requirement.

Confidence in this Review

3-Expert (read the paper in detail, know the area, quite certain of my opinion)


Reviewer 5

Summary

In this paper, they consider the problem min f(g(x)) + R(x) where f is sum f_i and g = sum g_i. They assume that 1) we can sample f_i and g_i. 2) f, g, R has bounded gradient 3) both the gradient of f(g(x)), f and g are Lipschitz 4) function value is bounded They show how to "minimize" them in rate T^(-4/9). (If the problem is not convex, they measure the progress by the expected norm of gradient instead of function value.) The previous best is T^(-2/7).

Qualitative Assessment

The result is quite good, especially the fact that it can match the optimal result for the case g or f is linear! Here are few comments: The author said they proposed the first stochastic proximal gradient method. However, their algorithm is almost the same as Wang et al [2016] except replacing the projection step by a proximal step. I think this small changes does not really qualify as a new algorithm. I would rather say this gives a new analysis of Wang et al [2016] algorithm. (By the way, I don't know Wang et al.) Wang et al [2016] assumed that the 4th norm of grad f and grad g is bounded in expectation. This is not really a "fourth order gradient". (probably a typo?) In your result, you assume grad f and grad g is bounded almost surely. So, it is a stronger assumption instead of a weaker assumption. (i.e. your result not just assumed fourth "order" (moment) gradient, but also kth "order" (moment) gradient for all k. By the way, why assumption 4(2) and (3) is written differently? both (2) and (3) is about the Lipschitz gradient of f and g.

Confidence in this Review

2-Confident (read it all; understood it all reasonably well)


Reviewer 6

Summary

In this paper authors proposed stochastic gradient method for composition optimization problem. In a special case this problem is classical empirical loss minimization problem. The work can be seen as generalization of a work of Wang [2016], however, in this paper they have stochastic algorithm, where in each iteration just 3 random samples are drawn and corresponding gradients/function values are computed. Authors also motivated their work by Reinforcement learning and showed some numerical experiments on this problem.

Qualitative Assessment

I like the extension of Wang [2016] to stochastic algorithm. However, there are few things I do not like 1. all those interesting quantities are assumed to be O(1) and hence the Theorems are not enough informative, to see how, e.g. variance of stochastic gradient is influencing the convergence. I think it would be much better if they use some constants to make it more informative. 2. there is a small typo in (5), the "1-\alpha" should be "\alpha-1" otherwise the discussion after theorem would not be valid :) 3. Assumption 3: there is twice "\forall x \forall x" 4. Line 171: why "without any assumption", what do you mean by that? there is quite a lot of strict assumptions (assumptions 1-4) or even equation (8).

Confidence in this Review

2-Confident (read it all; understood it all reasonably well)